# Human Papillomavirus Perceptions, Vaccine Uptake, and Sexual Risk Factors in Students Attending a Large Public Midwestern University

**DOI:** 10.3390/vaccines12060671

**Published:** 2024-06-17

**Authors:** JaNiese E. Jensen, Linder H. Wendt, Joseph C. Spring, Jay Brooks Jackson

**Affiliations:** 1Department of Pathology, Carver College of Medicine, University of Iowa, Iowa City, IA 52242, USA; brooks-jackson@uiowa.edu; 2Institute for Clinical and Translational Science, University of Iowa, Iowa City, IA 52242, USA

**Keywords:** human papillomavirus, vaccinations, university students

## Abstract

*Background:* It was to understand HPV vaccination patterns, uptake, perceptions, and sexual risk factors in students at a Midwest public university. *Participants:* Students were enrolled during the spring 2024 semester at the University of Iowa. *Methods:* A survey was developed and emailed to 28,095 students asking demographic, general and sexual health, and HPV-related questions. *Results:* The response rate was 4.9%, with 76% females and a median age of 22. The HPV vaccine uptake was 82%, with 88% recommending the vaccine. Parental preference was the main reason for being unvaccinated. The median age of sexual debut was 17 years, with a median of 2 sexual partners. Vaccination was associated with female, health science, sexually active, and COVID-19/influenza vaccinated students. *Conclusions:* HPV vaccine uptake at University of Iowa students is higher than the national and Iowa averages. Increased education regarding HPV vaccination is still needed, particularly in males, those not having sex, and those not receiving other vaccines.

## 1. Introduction

The most common sexually transmitted infection (STI) in the United States (US) is the human papillomavirus (HPV) [1]. HPV is a double stranded DNA virus that typically affects cutaneous or mucosal surfaces. HPV is most frequently spread through skin––skin or skin––mucosal sexual contact [2]. Common sites of infection from sexual contact include anal, vaginal, penile, cervical, and oropharyngeal epithelium [3]. HPV is a circular genome that encodes eight proteins, including six regulatory regions called early regions, E1, E2, E4, E5, E6, E7, and two capsid regions called late regions, L1 and L2, which each cause increased replication and spread of the virus [4]. The Centers for Disease Control and Prevention (CDC) estimates that 13 million people will be diagnosed with a new HPV infection each year, with 42.5 million people infected at any given time in the US alone [1]. HPV imposes a significant burden on the US healthcare system, as it is attributed to around USD 775 million in direct medical costs, which is higher than medical spending on all other STIs except the human immunodeficiency virus (HIV) [1].

There are over 200 types of HPV strains, classified as low and high-risk, that cause a range of diseases [5]. Although HPV infection is often asymptomatic, low-risk strains, such as HPV 6 and 11, can result in nonmalignant lesions, such as genital warts and respiratory papillomatosis, and high-risk strains, such as HPV 16, 18, 31, and 33, can result in neoplasia, with HPV 16 accounting for approximately of 55% of cervical cancers [5]. More than 90% of cervical and anal cancers, 70% of vaginal, vulvar, and oropharyngeal cancers, and 60% of penile cancers are estimated to be associated with an HPV infection [6].

Due to adolescents and young adults being at increased risk of sexually acquired HPV, prevention prior to sexual debut is necessary. The main and most effective method of HPV prevention is through HPV vaccination. Other methods to decrease the spread of HPV include condom use, abstinence, and fewer sexual partners [7]. There are three major types of HPV vaccines available in the world; however, only the Gardasil 9^®^ vaccine is currently in use in the United States [8]. The nonavalent Gardasil^®^ vaccine protects against HPV 6, 11, 16, 18, 31, 33, 45, 52, and 58. The bivalent vaccine protects against HPV 16 and 18, and the quadrivalent vaccine protects against HPV 6, 11, 16, and 18 [8]. The CDC currently recommends all children aged 11–12 years receive two doses of the vaccine 6–12 months apart [9]. For those 15–26 years old, three doses of the vaccine should be given. However, vaccination can begin as early as 9 years old and as late as 45 years old, based on risk assessment. Many governing bodies have set goals to increase HPV vaccination rates. The US Healthy People 2023 goal is for 80% of adolescents to have completed the vaccine series [10]. As of 2022, the US has fallen short of this goal.Only 76% of adolescents aged 13–17 years had received at least one dose of the vaccine, and 62.6% had received the complete series [11]. In Iowa, the rate is even lower, with 45.2% of adolescents, 44.1% male and 47.5% female, aged 13–15 years, having completed the series [12].

With the HPV vaccination catch-up period ending at 26 years of age for most healthy individuals and the increased risk of HPV infection in youth and adolescents, understanding HPV perceptions and vaccination trends among college-aged individuals is important. To our knowledge, there have been no studies on HPV vaccine uptake and perceptions in large Midwest universities. Similar studies in the US in the South and Mid-Atlantic have found HPV vaccine uptake and HPV knowledge to be low in university students [13,14]. The objective of this study was to determine the uptake and perceptions of the HPV vaccine in students enrolled at the University of Iowa. The secondary purpose of the study was to determine the sexual risk factors of students and the associated impact on their perception of HPV infection risk. It was hypothesized that HPV vaccination rates would be higher than the national average and that sexual risk factors would be similar to those found in other university students.

## 2. Materials and Methods

A survey consisting of 16 questions was created by study investigators. The survey asked questions regarding demographics, health history (including sexual history), and HPV-specific questions. Of the 29,908 students currently enrolled at the University of Iowa, emails were obtained through the Office of the Registrar for 28,095 students who had their email available through the student directory. Those who were not emailed had blocked their email from being publicly used, which is in line with the Family Educational Rights and Privacy Act. The survey was then emailed to students in April 2024 with a unique link through REDCap^®^ that could be completed on a phone or computer.

Students were eligible if they were currently a student at the University of Iowa with access to an email and 18 years or older. Students were required to answer the first question of “age” for eligibility verification but could choose to answer as many of the other questions as they desired. Students were told that consent was assumed to be granted based on the completion of the survey.

Students submitting the survey were eligible to win one of three $1000 scholarships applied to tuition through a raffle of survey submissions. Participants had the option to opt into this raffle after they submitted the survey, and responses to the raffle were collected through a secondary REDCap^®^ survey that was not linked to their primary survey answers. The number of questions completed by each participant did not influence their ability to be entered into the raffle drawing. Variables were summarized using counts and percentages for categorical data and using medians and interquartile ranges for continuous data. Fisher’s exact test was used to evaluate differences in categorical data across strata, while Wilcoxon rank sum tests were used for continuous data. All data analysis was performed using R, version 4.3.3, in R studio [15,16,17]. Demographic data from the survey were compared to the University of Iowa registrar’s profile of currently enrolled students for spring 2024. The study was conducted in accordance with the Declaration of Helsinki, and Institutional Review Board (IRB) approval was granted through the University of Iowa (IRB #202403543).

## 3. Results

### 3.1. Survey Responses—Demographics

Of the 28,095 students emailed, 1380 responded beyond the first survey question of “age”, with a total percentage of completion of 4.9% of the emailed students. The demographics are described by each question in Table 1. The median age was 22 (IQR 20, 25), with a range of 18 to 67. Females comprised 76% of those surveyed, and 24% were male. There were no responses for intersex. The largest percentage of students were enrolled in the College of Liberal Arts and Sciences at 44%, with the nine other college options being combined into a “health science colleges” response of 33% and an “other” response of 23%. The most frequently selected sexual identity response was straight (73%) followed by bisexual (17%), gay (3.3%), and lesbian (2.6%). Free responses to “other” for sexual orientation/identity included “demisexual”, “queer”, “pansexual”, and “questioning”. Democrat was the most common political party reported (54%). Over half of the students (56%) were considered undergraduates.

### 3.2. Survey Responses—Health History

General health with sexual history responses aredescribed in Table 2. Students most frequently received healthcare at a hometown clinic (56%) and student health (52%). Most students had received a COVID-19 vaccination at some point (93%), and 67% had received the annual influenza vaccine. For the sexual activity questions, sex was defined as vaginal, anal, or oral sexual activity. If the student responded yes to “have you ever had sex”, three more questions populated. If they said no, these questions were not asked. Most students have had sex (81%). The average age of sexual debut was 17 years (IQR 16, 19). The average number of sexual partners was two (IQR 1, 6) with a range of 0 to 150. The question asking if they have received a Pap test/smear was only asked of those who responded as female and 21 or older, of whom 72% had received a Pap test.

### 3.3. Survey Responses—HPV Specifics

Questions related to HPV vaccination and knowledge are described in Table 3, and 94% of students had heard of HPV prior to the survey. Of those surveyed, 82% had received the HPV vaccine. Students were not asked which type of HPV vaccine they received. The median age for HPV vaccination was 14, with an IQR of 12, 16. Only 41% of students correctly selected breast cancer as the disease not associated with HPV, with respiratory infection as the next highest selected response (34%). The top reasons for why students had not received the HPV vaccine were related to parental preference (22%), lack of knowledge of the vaccine (20%), or vaccine was not offered (19%). The responses to “other” for why the student did not receive the vaccine included being too old for vaccination, being sexually monogamous, lack of motivation to receive vaccination, prior diagnosis of HPV, fear of vaccinations, and dislike of ingredients in the vaccine.

### 3.4. HPV Vaccine Associations

A comparison of survey responses of students who were vaccinated or unvaccinated is shown in Table 4. Those who were HPV-vaccinated were significantly more likely to have been vaccinated for COVID-19 (*p* < 0.001) and annual influenza (*p* < 0.001). Students who considered themselves Democrat were more likely to receive the vaccine (*p* < 0.001). Vaccinated students reported to have engaged in sex significantly more (*p* = 0.019) and while significantly younger (*p* = 0.038) compared to those who were unvaccinated. The age of first sexual encounter differed between vaccinated students at 17 years (IQR 16, 19) compared to unvaccinated students at 18 years (IQR 16, 19). The average number of sexual partners was significantly different between vaccinated and unvaccinated at two (IQR 1, 6) and one (IQR 0, 4), respectively (*p* < 0.001). Vaccinated females who were 21 years or older were significantly more likely to have had a Pap test than unvaccinated females (*p* = 0.005). HPV-vaccinated individuals would recommend the HPV vaccine more frequently than unvaccinated students (*p* < 0.001). Healthcare was utilized significantly more by those who were vaccinated in the past year..Fifty-three percent of vaccinated and 45% of unvaccinated students had used student health (*p* = 0.04), and 41% of vaccinated and 29% of unvaccinated students had used UIHC other than student health (*p* < 0.001). There was no significant difference between graduate or undergraduate students, history of STI, in-state status, or sexual identity on vaccination rates.

When comparing sexual debut to HPV vaccination, there was strong evidence that it is more common to have the HPV vaccine prior to having sex (*p* < 0.0001). In comparing those who had the HPV vaccine before sex and those who had sex before the HPV vaccine, 624 (78%) had the HPV vaccine first, while 174 (22%) had sex first. The relationshipbetween sexual debut and HPV vaccination is shown in Figure 1.

### 3.5. HPV Knowledge Associations

Students enrolled in the College of Liberal Arts and Sciences (CLAS) correctly selected breast cancer as the disease not associated with HPV at a rate of 36% and respiratory infection at a rate of 35%. Students enrolled in a health-science college selected breast cancer 44% of the time, and 37% selected respiratory infection. We have significant evidence that these two student groups were different in their rate of selecting breast cancer (*p* = 0.009), but no evidence that they were different in their rate of selecting respiratory infection (*p* = 0.515). In response to the most effective way to prevent HPV infection, 58% of students selected vaccination, and 41% selected condom use. For students in health sciences, 66% selected vaccination, and 33% selected condom use. We have strong evidence that these two student groups were different in their rate of selecting both vaccination (*p* = 0.005) and condom use (*p* = 0.007).

### 3.6. Gender Associations

A comparison of select survey responses to gender is shown in Table 5. Complete survey responses by gender are shown in Table A2 in Appendix A. There was a significant difference in males and females in response to age (*p* = 0.011), healthcare utilization at a clinic in their hometown (*p* < 0.001), or QuickCare/Urgent Care (*p* < 0.001), if they have had sex (*p* = 0.006), if they have heard of HPV (*p* = 0.002), if they have been vaccinated for HPV (*p* < 0.001), and when they received the first dose (*p* < 0.001). The median age for males was 23 (IQR 20, 26) and females was 22 (IQR 20, 25). Males had received their first dose of the HPV vaccine at a median age of 15 (IQR 12, 18) and females at 13 (IQR 12, 16).

### 3.7. Sexual Activity Associations

The sexual activity associations with survey responses are shown in Table 6. A comparison of those who have and have not had sex, with gender and sexual orientation, can be seen in Figure 2. There was a significant difference between sex assigned at birth (*p* = 0.006) and sexual identity (*p* < 0.001) in those who have had sex and those who have not. There was a significant difference between the number of sexual partners in those who had been diagnosed with an STI and those who had not (*p* < 0.001). IQR was used in reporting this data, as the data was not normally distributed. There was also a significant difference in the number of sexual partners based on sexual identity (*p* < 0.001), with the highest number of sexual partners in gay students (n = 6) and the lowest number in lesbian and asexual students (n = 1). The range of sexual partners for gay students was 0 to 150, for lesbian students (0, 50), asexual students (0, 25), straight (0, 137), bisexual (0, 50), and other (0, 23). Overall, as age increases, so does the number of students who are sexually active. Students aged 18 years were sexually active at a rate of 67%, and this increased steadily each year. As seen in Figure 3, when the rate of sexual activity is stratified by sex assigned at birth, males were more frequently sexually active in each age group. By age 32, 100% of respondents had been sexually active regardless of sex assigned at birth, with an exception at age 47 with 50% of 47-year-old students having had sex (47 years, n = 2).

## 4. Discussion

### 4.1. HPV Vaccine Uptake and Perceptions

Those aged 21–24 years old have the highest rate of infection with human papillomavirus than any other age group [18]. This age group also has an increased risk of infection due to the risk associated with sexual activity. Therefore, understanding these risk factors and how HPV infection is perceived byby university students is important, as this group is the ideal target for HPV vaccination catch-up, which can occur up to 26 years of age.

Females (*p* < 0.001), health-science students (*p* = 0.007), those who had sex (*p* = 0.019), and those who had received their COVID-19 (*p* < 0.001) or annual influenza (*p* < 0.001) vaccines were more likely to be vaccinated for HPV, as seen in this study. The overall uptake of the HPV vaccine was found to be 82% in the student population, which is higher than the average vaccination rates in Iowa and in similar studies of university students. In a 2022 study of HPV vaccination rates in states, Iowa had the fourth highest uptake among states, with 74.9% of adolescents aged 13–17 years having received the HPV vaccine [19]. In a 2023 study of 417 Mid-South college students aged 18–26 years, 48.7% of students had been vaccinated for HPV, and 53% of females had received at least one dose of the vaccine [13]. The increased rate of vaccination of students surveyed in this study may be explained by Iowa having a higher uptake of the vaccine than many other states; however, the surveyed students had a higher vaccination rate than the Iowa average as well. It is postulated that this may be due to an increase in vaccination knowledge and a push for vaccination that has been occurring in the US recently. This higher rate may also be explained by the higher number of females, as well as health-science students who responded to the survey, as it is expected that health-science students may have higher vaccination rates overall.

Female, vaccinated, and health-science students were more knowledgeable about HPV infection than other groups based on responses to HPV knowledge questions. A 2009 study showed that 93.6% of participants had heard of HPV, which is similar to the 94% found in this survey [20]. Other studies have also shown that females know about HPV significantly more than men, which was shown in this survey [21,22]. This difference may be because many females will receive cervical cancer screening in their lifetime, so discussions around HPV-associated cervical cancer may happen more frequently in healthcare or personal settings for females. Vaccinated students selected HPV vaccination as the best method for HPV prevention significantly more than unvaccinated students (*p* = 0.005). The correctness of response is similar to a study of students at Villanova, in which vaccinated students answered the HPV knowledge questions correctly more frequently than unvaccinated students with a near-marginal significance (*p* = 0.12) [14]. Vaccination was selected by 66% of health-science students as the best method to prevent HPV, compared to 58% of College of Liberal Arts students, and this difference was statistically significant (*p* = 0.005).

In assessing the level of knowledge of HPV, the survey responses to the question “which of the following diseases does HPV not cause” showed that there may have been confusion regarding the response options given to the participants, regardless of vaccination status. With 41% of students selecting the correct answer of “breast cancer”, it can be extrapolated that 59% of students did not know that HPV was associated with some of the other listed diseases. The next most selected disease was respiratory infection at 34%. The high response rate for respiratory infection is likely explained by what the students view as respiratory infections. The survey was designed for respiratory infection to imply respiratory papillomatosis, which is a known HPV-associated disease, without explicitly stating respiratory papillomatosis. This wording could have led to confusionamong students if they viewed respiratory infection as the colloquially used definition of upper or lower respiratory infection, such as a common cold, influenza, or pneumonia. With 37% of students enrolled in a healthcare college choosing respiratory infection, it is likely that the use of “respiratory infection” caused confusion.

Understanding why individuals were not vaccinated is important in the young adult population, as increased education may increase vaccination. In those who were not vaccinated, the most common reasons as to why they were not vaccinated were parental preference, they did not know about the vaccine, or the vaccine was never offered to them. Parental preference can be addressed by educating college students at their yearly healthcare visits now that they are able to make their own decisions on healthcare. It is also important to continue to offer HPV vaccines up to age 26, which may address the knowledge gap or lack of access to the vaccine. Unvaccinated males also did not know about the vaccine, at a rate of 29% compared to 14% of unvaccinated females. Education should also be targeted toward males, as males were vaccinated later than females and were offered the vaccine at a lower rate than females. Unvaccinated males also did not know about the vaccine more frequently than unvaccinated females. With males being part of the recommended group to be vaccinated, increasing education and opportunities to become vaccinated is necessary to increase overall uptake.

Twelve percent of unvaccinated students reportedly did not receive the vaccine because they were not sexually active; however, 0% of unvaccinated students were concerned about the vaccine promoting sexual activity. This is important, as the people who are not sexually active yet are the ideal population to vaccinate for HPV. In a 2024 study by Palmer et al. in Scotland, if the HPV vaccine was given to 12–13-year-old girls prior to sexual debut, the vaccine was found to prevent cervical cancer at a rate of 100% [23]. There was strong evidence to support the idea that vaccination rates were higher in students prior to sexual debut (*p* < 0.0001) than those after sexual debut. This finding is reflected in the fact that the median age of vaccination in the student population was 14, and the average age of sexual debut was 17. University of Iowa students are being protected from HPV infection and subsequent disease by being vaccinated prior to their sexual debut. Therefore, it is necessary to increase education for those who are unvaccinated, particularly those who have not been sexually active, as these students have the greatest potential for the benefits associated with vaccination.

A secondary association found in this study was that healthcare is being used at a significantly higher rate byby those who were vaccinated compared to those who were not vaccinated. Males were more likely to have not utilized healthcare in the last year than females (*p* = 0.004). A study in Brazilon HPV knowledge showed that young adult females were more likely to have visited primary care in the last year [24]. Engaging in healthcare at a higher rate also could increase knowledge of and access to HPV vaccinations, which may explain why females were vaccinated at a higher rate than males.

It is important to note that HPV has a major impact not only in the US but worldwide. Increasing education in the US and beyond is important. Education should be targeted towards similar groups of people as found in this study, including those not having sex, males, and uneducated individuals. However, education may need to be tailored differently in other countries and should focus on targeting vaccine mistrust, addressing the costs of vaccines, and creating equitable access to vaccines [25].

### 4.2. Sexual Activity in College Students

Because HPV is most frequently spread through sexual contact, it is important to understand sexual risk factors. Of the University of Iowa students, 81 out of 100 students have had sex with a median number of two lifetime sexual partners. When sex is defined as vaginal intercourse between opposite-sex partners, 40.5% of females and 38.7% of males aged 15–19 years in the US have ever had sex [26]. When sex is defined as vaginal, oral, or anal sex, 41% of females and 39% of males aged 15–17 years and 71% of both males and females aged 18–19 years have had sex [27]. In a 2023 student wellness survey conducted at the University of Iowa, undergraduates had an average of two sexual partners in the last 12 months, while graduate students had 1.5 [28]. In a study of lifetime partners in secondary school adolescents, the average number of lifetime partners was 1.5 if the sexual debut was within 12 months and 4 if the sexual debut was greater than 35 months prior to the survey [29]. A College Stats survey of current and graduated college students showed that the average number of sexual partners in collegewas 4.98 in men and 4.90 in women [30]. This study also noted that the average number of lifetime partners for men was 14.22, and for women, it was 11.41 [30].

The risk of HPV infection can be related to an increased number of sexual partners and having another STI. A study of women aged 35–60 years found that, for women with five or more lifetime sexual partners, the rate of high-risk HPV infection increased two-fold [31]. Of students who have had sex, 11% have had a STI. This question was only asked to those who answered yes to having had sex. Almost one-half of the STI incidence in the US is in people aged 15 to 24 years, with 7.1 million having acquired a disease-associated HPV infection in 2018 [32]. A University of Iowa wellness survey found that 13.5% of undergraduate and 33.4% of graduate students have been tested for HIV in their lifetime [28]. The Iowa incidence of STIs per year is 457.2 per 100,000 people [33].

### 4.3. Advantages, Limitations, and Future Studies

An advantage of completing a survey is that nearly all students had the opportunity to participate, decreasing the sampling bias. Like similar studies before, a survey was used to allow for greater access and ease of completion of the survey. An advantage of this study is that both undergraduate and graduate students were included, allowing for a larger population to be surveyed and an increased inclusion of individuals who fall into the catch-up vaccination age range and may have finished undergraduate studies.

Although the information presented in this survey discusses young adults in a university setting, the recent change in age recommendations for the HPV vaccine is an important point to consider when discussing these results. The HPV vaccine became available in 2006 in the US for females aged 9 to 26 years and was expanded to include males aged 9 to 21 years in 2011 [34]. However, it was not until 2019 that males up to 26 years of age were recommended for catch-up vaccination. With the median age of the population surveyed being 22, most participants would fall into this recommendation; however, with the age range being 18 to 67 years, vaccination may not have been available to all who responded.

Another point to address is the demographics of those who responded to the survey do not entirely represent the same demographics of the University of Iowa student population. According to the office of Iowa registrar’s office, the spring 2024 enrollment was 29,908 with 55.1% of students identifying as female [35]. The survey had a higher response rate for females, at 76%. A key differentiating factor is that the survey asked specifically for sex assigned at birth, while the censusreports “sex” without further definition. In comparing colleges, an expected increase in interest from those students in health-science colleges was seen. An example of this can be seen with 15% of Carver College of Medicine students responding to the survey when the medical school makes up only 3% of the student body. However, the college with the largest enrollment is the College of Liberal Arts and Sciences, with 45% of the student population, which was similar to the survey response of 44%. The number of Iowa resident students was 56.6%, which was lower than the survey response of 68%. Students identified as heterosexual at a rate of 73%, which is similar to the 2021 youth survey conducted by the CDC that found 74.2% of youth identified as heterosexual [36].

The differences seen in the survey and university populations can likely be explained by a few factors. The firstis that the survey had a 4.9% response rate, and it may not be representative of the student body as a whole. Second, students in healthcare fields likely have more interest in the results of the study. Thus, they may have completed the survey at a higher rate. Third, as stated earlier, Iowa is the fourth most vaccinated state in the country. Therefore, Iowa students may have an increased knowledge of or interest in HPV, which may increase the response rate. Finally, the difference in male and female response rates is likely explained by HPV infections being commonly known to cause cervical cancer, which is a disease that affects females. Males can be impacted by HPV-associated disease, but they are not being screened for these diseases regularly like females are with Pap tests. In general, other surveys have also shown that females respond at an increased rate to surveys regarding HPV when compared to males [20,24].

Another limitationof this study is that only University of Iowa students were surveyed making these data potentially not generalizable to other institutions. This survey had a mostly female, in-state Iowa resident, Democrat, and straight (heterosexual) student response. It is likely that universities in different geographic locations outside of the Midwest may have different response rates to the questions presented in this survey.

A future research direction would be to initiate operational research studies to reach more university students, with a focus on students considered at high risk for infection and groups with the lowest vaccination rates. The high-risk groups include those who have sex with multiple partners, those who have a combination of oral, anal, and/or vaginal sex, and men who have sex with men, as testing for HPV is not common for men. The lowest vaccination rate groups include males, those who choose not to receive other vaccines, those in colleges other than health sciences, and those who are not having sex. Another group to focus educational efforts on would be the parents of adolescents, as some students reported that they were unvaccinated due to parental preference.

## 5. Conclusions

HPV vaccination rates were higher than the US national average at the University of Iowa, and the students had an overall positive perception of HPV vaccines. Female, health science, and sexually active students are more likely to be vaccinated than other groups. Students more often than not received the HPV vaccine prior to sexual debut. With HPV vaccination being available up to age 26 in the US, it is necessary to increase the education of males, those not sexually active, those with sexual risk factors, and those studying outside of the health sciences to ensure adequate protection from HPV through vaccination.

## Figures and Tables

**Figure 1 vaccines-12-00671-f001:**
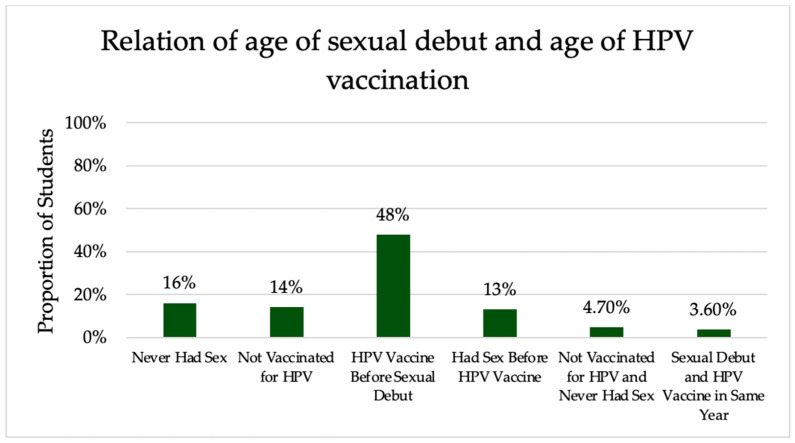
Comparison of sexual debut and HPV vaccination.

**Figure 2 vaccines-12-00671-f002:**
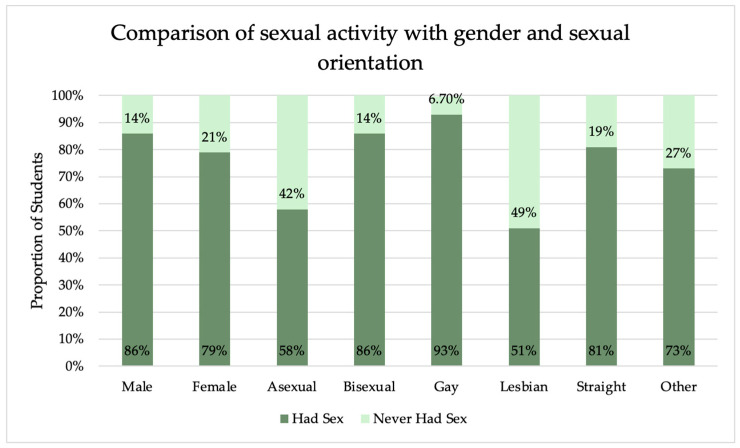
Comparison of those who have and have not had sex and their sexual orientation and sex assigned at birth.

**Figure 3 vaccines-12-00671-f003:**
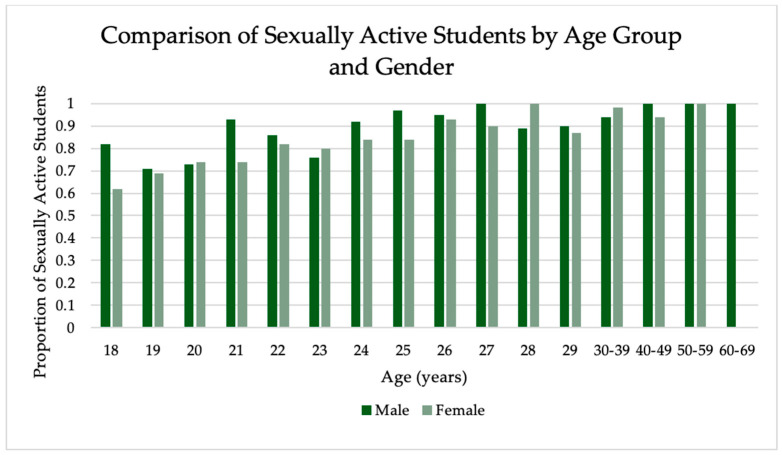
Proportion of sexually active students based on current age by year from 18–29 years then by decade for 30, 40, 50, and 60 years and sex assigned at birth.

**Table 1 vaccines-12-00671-t001:** Demographics of survey respondents.

	N = 1380 (%)
**Age, median**	22
**Sex assigned at birth**	
Male	331 (24%)
Female	1044 (76%)
**In which college are you currently enrolled?**	
College of Liberal Arts and Sciences	602 (44%)
Medicine/Dentistry/Public Health/Nursing/Pharmacy	454 (33%)
Other	321 (23%)
What is your current level in school?	
Undergraduate	768 (56%)
Graduate	600 (44%)
Other	5 (0.4%)
**Are you considered an in-state resident of Iowa for tuition purposes?**	925 (68%)
**Sexual Identity/Sexual Orientation**	
Straight (heterosexual)	1006 (73%)
Bisexual	234 (17%)
Gay or Lesbian	81 (5.9%)
Asexual or Other	54 (3.9%)
**Which political party do you most closely align with?**	
Democrat	735 (54%)
Independent	313 (23%)
Republican	213 (16%)
Libertarian or Other	99 (7.3%)

**Table 2 vaccines-12-00671-t002:** Descriptive responses of students to the “General Health Questions” portion of the survey.

	N = 1380 (%)
**Where have you utilized healthcare services in the last year? Select all that may apply.**	
Clinic located in your hometown	772 (56%)
University of Iowa Student Health	713 (52%)
QuickCare/Urgent Care/Emergency Department	607 (44%)
UIHC—Other than student health	531 (38%)
Other Iowa City clinic	113 (8.2%)
**Have not used healthcare services in the last year**	50 (3.6%)
**Did you receive the COVID-19 vaccination at any point?**	1279 (93%)
**Did you receive the annual influenza vaccination in 2023–2024?**	911 (67%)
**Have you ever had sex? Sex is defined as vaginal, anal, or oral sexual activity.**	1105 (81%)
**If yes, at what age did you have sex for the first time? (years, median)**	17
**If yes, how many sexual partners have you had? (median)**	2
**If yes, have you ever been diagnosed with a sexually transmitted infection (ex: HIV, syphilis, gonorrhea, chlamydia, genital warts)?**	122 (11%)
**If female and 21 or older: Have you ever received a Pap test/smear?**	474 (72%)

**Table 3 vaccines-12-00671-t003:** Descriptive responses of students on the “HPV Specifics” portion of the survey.

	N = 1380 (%)
**Have you heard of the human papillomavirus (HPV) prior to this survey?**	1287 (94%)
**Which of the following diseases does HPV NOT cause? Select one best answer.**	
Breast cancer	555 (41%)
Respiratory infection	458 (34%)
Plantar or genital warts	234 (17.1%)
Oropharyngeal, cervical, or anal cancer	111 (8.1%)
**Have you received the HPV vaccine (e.g., Gardasil)?**	1126 (82%)
**How old were you when you received your first dose?**	14
**Why have you not received the vaccine? Select the single best reason.**	
My parent did not want me to get the vaccine	53 (22%)
I did not know about the vaccine	49 (20%)
The vaccine was never offered	46 (19%)
Not sexually active	29 (12%)
I did not want the vaccine	19 (7.8%)
Worried about a bad reaction or side effects	13 (5.3%)
I did not know where to get the vaccine	8 (3.3%)
Concerns about the promotion of sexual activity	0 (0%)
Other	27 (11%)
**Which is the most effective way to prevent HPV infection?**	
Condom use	536 (39%)
Vaccination	823 (60%)
Other	12 (1%)
**Would you recommend others get the HPV vaccine?**	
Yes	1208 (88%)
No	25 (1.8%)
Unsure	141 (10%)

**Table 4 vaccines-12-00671-t004:** Comparison of HPV vaccination status with survey responses and corresponding *p*-values.

	Received HPV Vaccine, N = 1126 (%)	Did Not Receive HPV Vaccine, N = 244 (%)	*p*-Value
Received COVID-19 Vaccine	1068 (84%)	201 (16%)	**<0.001**
Received annual influenza vaccination in 2023–2024	783 (87%)	122 (13%)	**<0.001**
Received Pap test/smear	414 (88%)	58 (12%)	**0.005**
Number of sexual partners	2 (1, 6)	1 (0, 4)	**<0.001**
Had sex	914 (83%)	182 (17%)	**0.019**
Recommend HPV vaccine			**<0.001**
Yes	1065 (89%)	137 (11%)	
No	9 (36%)	16 (64%)	
Unsure	49 (35%)	90 (65%)	
Most effective way to prevent HPV infection			**0.005**
Condom use	423 (79%)	110 (21%)	
Penicillin prophylaxis	1 (25%)	3 (75%)	
Spermicidal foam/cream	2 (100%)	0 (0%)	
Vaccination	690 (84%)	129 (16%)	
Hand washing	4 (67%)	2 (33%)	
Utilized University of Iowa Student Health in past year	596 (84%)	111 (16%)	**0.040**
Utilized UIHC other than student health in past year	458 (87%)	71 (13%)	**<0.001**
Grade Level			0.5
Undergraduate	634 (83%)	127 (17%)	
Graduate	485 (81%)	112 (19%)	
Other	4 (80%)	1 (20%)	
Sexual Identity/Sexual Orientation			0.10
Asexual	18 (75%)	6 (25%)	
Bisexual	205 (88%)	27 (12%)	
Gay	37 (84%)	7 (16%)	
Lesbian	31 (86%)	5 (14%)	
Straight (heterosexual)	807 (81%)	192 (19%)	
Other	25 (83%)	5 (17%)	
Political Party			**<0.001**
Democrat	642 (88%)	88 (12%)	
Independent	237 (76%)	74 (24%)	
Libertarian	23 (70%)	10 (30%)	
Republican	158 (75%)	53 (25%)	
Other	54 (82%)	12 (18%)	
Current College			**0.007**
College of Liberal Arts and Sciences	484 (81%)	113 (19%)	
Medicine/Dentistry/Public Health/Nursing/Pharmacy	392 (87%)	61 (13%)	
Other	248 (78%)	69 (22%)	
In-state resident	761 (83%)	159 (17%)	0.5
Age of Sexual Debut in years	17 (16, 19)	18 (16, 19)	**0.038**
Diagnosed with an STI	97 (81%)	23 (19%)	0.4

Bolded *p*-values indicate significance with *p* ≤ 0.05.

**Table 5 vaccines-12-00671-t005:** Select survey responses compared to male and female gender responses.

	Male, N = 331 (%)	Female, N = 1044 (%)	*p*-Value
Age	23	22	**0.011**
Had Utilized Healthcare			
Clinic located in your hometown	146 (44%)	622 (60%)	**<0.001**
University of Iowa Student Health	174 (53%)	538 (52%)	0.8
UIHC—Other than student health	117 (35%)	413 (40%)	0.2
Other Iowa City clinic	20 (6.0%)	92 (8.8%)	0.13
QuickCare/Urgent Care/Emergency Department	115 (35%)	491 (47%)	**<0.001**
Have not used healthcare services in the last year	21 (6.3%)	29 (2.8%)	**0.004**
Other	13 (3.9%)	35 (3.4%)	0.6
Had sex	282 (86%)	819 (79%)	**0.006**
Age of Sexual Debut	17 (16, 19)	17 (16, 19)	0.084
Had an STI	31 (11%)	91 (11%)	>0.9
Had heard of HPV	298 (90%)	985 (95%)	**0.002**
Received the HPV vaccine	239 (73%)	883 (85%)	**<0.001**
Age of first HPV dose	15	13	**<0.001**

Bolded *p*-values indicate significance with *p* ≤ 0.05.

**Table 6 vaccines-12-00671-t006:** Comparison of number of sexual partners to STI rates and sexual identity.

	Number of Sexual Partners N = 1380 (IQR)	Min	Max	*p*-Value
**Diagnosed with an STI**				**<0.001**
Yes	10 (5, 20)	1	150	
No	3 (1, 6)	1	137	
**Sexual Identity/Sexual Orientation**				**<0.001**
Asexual	1 (0, 2)	0	7	
Bisexual	4 (1, 10)	0	50	
Gay	6 (4, 18)	0	150	
Lesbian	1 (0, 3)	0	50	
Straight (heterosexual)	2 (1, 5)	0	137	
Other	3 (0, 10)	0	40	

Bolded *p*-values indicate significance with *p* ≤ 0.05.

## Data Availability

All research data can be found in Appendix A.

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
