# Peer review of "Human Papillomavirus Perceptions, Vaccine Uptake, and Sexual Risk Factors in Students Attending a Large Public Midwestern University"

_vaccines, 2024, doi:10.3390/vaccines12060671_

Round 1

Reviewer 1 Report

Comments and Suggestions for Authors

Human papillomavirus perceptions, vaccine uptake, and sexual risk factors in students attending a large public Midwest university.

Manuscript ID: vaccines-3041116

In the present project the authors developed a survey in order to explore perceptions and uptake of the HPV vaccine among a population of newly enrolled students at the University of Iowa, and to establish associations with sexual risk factors and demographic aspects. The authors found that although the uptake of HPV vaccine among participants was higher than the national average, more education and information about HPV vaccination is needed, in particular in males and those not having an active sexual life.

The manuscript is clearly written, the results are interesting and relevant due to importance and still high prevalence of HPV infection. However, the study is constrained to a local population, with particular characteristics that may limit the interest of the worldwide community. Thus, to improve the manuscript the authors should try to compare their results with those from other countries, and try also to use what they have found to help conducting better prevention and vaccination programmes in less favoured countries, where HPV is still a serious problem.

In the Discussion section the authors tend to repeat the results (figures, significance levels, etc) too many times. They should try to avoid reiteration and better concentrate in showing the meaning of their observations.

Figure 2 is not necessary, since it does not provide extra information. In fact, Figure 3 is much more informative.

Reviewer 2 Report

Comments and Suggestions for Authors

Comment about Objectives of the study. The objectives of the study must be expressed with clarity and brevity to facilitate future readers.

Comment about novelty of the study. The authors must also explain in detail the gaps in the literature that are covered by this particular study. Also, the authors must comment on the advantages of their approach and methodologies followed over those of previous similar studies performed previously.

Comment about controls. Please explain what type of controls you employed in this work. How did you calibrate the sample for diversity (gender, race, sexual orientation, professional qualifications etc.)?

The inclusion of all the tables in the main text provides details regarding the data, but on the other hand, it makes the manuscript long and boresome. I suggest to move them to supplementary material or to appendices in the revised submission.

Please include a larger number of graphs to show the results in a summary form.

The Discussion should be divided in three sub-sections for easier flow of reading.

The references are OK.

Conclusions. The conclusions do not give a take-home message with the results of the study. It should be rewritten to bring in line with the findings of the study.

Overall. Improvements are necessary before possible acceptance and re-reviewed after revision.

Reviewer 3 Report

Comments and Suggestions for Authors

This study made a questionnaire about HPV vaccination to students. The manuscript is easy to read and to understand. The manuscript could benefit from more technical description of the type of vaccine, mechanism of action, efficacy, secondary as well as HPV virus description. Gender identity definitions could be described in more detail. It looks like a high percentage of non-straight responders in comparison to other populations.

Additional comments:

(1) In the Introduction. Could you please show the structure of HPV and how it infects the target cells (the pathogenesis) in a Figure?

(2) In the Introduction, could you please make a table showing the different types of HPV, and their oncologic risk?

(3) In the Introduction, could you please describe the 3 major types of HPV vaccines, indications and contraindications (secondary effects).

(4) In Figure 1, section 6. What exactly does “asexual” sexual orientation mean? Are the different definitions present in a reference publication such as DSM?

(5) Line 87. Could you please add the R packages and their reference/internet website link?

(6) Was R studio used?

(7) Should Declaration of Helsinki be cited in line 90?

(8) In Table 1. Please add “median” after “age” in the first row.

(9) Is the fact that 76% are female some kind of bias?

(10) A 26% of people who responded the survey were not heterosexual. Why so high? (One would expect around 1-3%)

(11) Line 103. Please define “demisexual, queer, pansexual, and questioning”.

(12) Line 104. I am not sure if 54% can be “majority” of cases.

(13) Table 2. Regarding “If yes, at what age did you have sex for the first time?” Is this the average? Same with variable “number of partners”.

(14) What percentage of student was vaccinated against HPV? Any difference between sex a vaccination? Should all result of the study be stratified according to the sex?

(15) Are there questions with missing data?

(16) In Table 4. Is it possible to calculate the odds ratio?

(17) Are the differences of the replies regarding HPV knowledge associations of section 3.5 statistically significant?

(18) Could you please explain why IQR was favored against STD?

(19) Did all student receive the same vaccine?

(20) What is the efficacy of the vaccine? What subtypes protect from?

(21) How students should prevent HPV infection independently from the vaccination?

Round 2

Reviewer 2 Report

Comments and Suggestions for Authors

The authors have responded to all points raised and have made extensive changes to improve the manuscript.